# Expression and role of melatonin membrane receptors in the hypothalamic-pituitary-testicular axis of Tibetan sheep in a plateau pastoral area

Dapeng Yang[1], Ligang Yuan[1,2]*, Guojuan Chen[3], Shaoyu Chen[1], Xiaojie Ma[1], Yindi Xing[1], Juanjuan Song[1]

1 College of Veterinary Medicine, Gansu Agricultural University, Lanzhou, China, 2 Gansu Key Laboratory of Animal Generational Physiology and Reproductive Regulation, Lanzhou, China, 3 Huangzhong District Animal Disease Control Center of Xining City, Xining City, Qinghai Province, China

* yuan2918@126.com

**Data Availability Statement:** All relevant data are within the manuscript and its Supporting information files.

## Abstract

MTNR1A and MTNR1B, two high-affinity MT membrane receptors found in mammals, mediate the activity of MT on the HPGA to regulate animal reproduction. Nevertheless, the expression patterns and function of the MTNR1A and MTNR1B genes in the HPTA of seasonal estrus sheep and perennial estrus sheep have not been elucidated. We studied the expression of MTNR1A and MTNR1B in the hypothalamic-pituitary-testicular axis (HPTA) of Tibetan sheep at different reproductive stages using histochemistry, enzyme linked immunosorbent assay (ELSIA), scanning electron microscopy, transmission electron microscopy, quantitative Real-time PCR (qRT-PCR), and Western blot (WB), and analyzed the relationship between their expression and reproductive hormone receptors. We also compared relevant characteristics between seasonal Tibetan sheep and non-seasonal Small Tail Han sheep in the same pastoral area. The results showed that MTNR1A and MTNR1B were expressed in all tissues of the Tibetan sheep HPTA, and both were co-expressed in the cytoplasm of epididymis basal and halo cells located at common sites of the epididymis basement membrane, forming an immune barrier. The qRT-PCR analysis showed that not only MTNR1A but also N-acetyltransferase (AANAT), hydroxyindole-oxygen- methyltransferase (HIOMT), androgen receptor (AR), and estrogen receptor α (ERα) mRNA expression was significantly upregulated in the testis and epididymis of Tibetan sheep during the breeding season, whereas no clear upregulation of these genes was observed in the tissues of Small Tail Han sheep. MTNR1A and MTNR1B are important regulators of the HPTA in sheep. MTNR1A mediates seasonal estrus regulation in Tibetan sheep. Both MTNR1A and MTNR1B may play important roles in formation of the blood-epididymal barrier. The results of this study should help advance research on the mechanism of reproductive regulation of the HPTA in male animals and provide reference data for improving the reproductive rate of seasonal breeding animals.

**Funding:** This study was financially supported by the Natural Science Foundation of Gansu Province, China in the form of a grant (23JRRA1420) received by DY. This study was also financially supported by the Fund of the Discipline Team Project of Gansu Agricultural University in the form of a grant (GAU-XKTD-2023) received by DY. This study was also financially supported by the Project of "Innovation Star" for Excellent Graduate Students in Gansu Province in the form of a grant (2021CXZX-357) received by DY. This study was also financially supported by the Gansu Agricultural University College Students Scientific Research Training Program in the form of a grant (20180335) received by DY. The funders had no role in study design, data collection and analysis, decision to publish, or preparation of the manuscript.

**Competing interests:** The authors declare no conflict of interest.

## 1. Introduction

Melatonin (MT) is an amine hormone synthesized in the pineal gland from 5-hydroxytryptophan in a reaction catalyzed by a series of aminotransferases and dehydrogenases. N-Acetyl aminotransferase (AANAT) and hydroxyindole-oxy-methyltransferase (HIOMT) are the main enzymes involved in catalyzing MT synthesis [1–3]. Similar to the mode of action of most hormones, MT requires receptor mediation to exert its effects in animal reproductive regulation [4]. Three subtypes of the melatonin receptor (MTNR) have been identified in mammals, MTNR1A, MTNR1B, and MTNR1C. In recent years, a fourth type of MTNR was identified in fish, known as MTNR1a-like, which can be classified as MTNR1D [5]. MTNR1A and MTNR1B are high-affinity MT membrane receptors found in mammals and mediate the multiple functions of MT in processes including sleep regulation [6], anti-inflammation processes [7], anti-tumor processes [8], and regulation of reproduction [9]. MTNR1A consists of 351 amino acids and is distributed in a variety of tissues and organs, including the brain, skin, testes, ovaries, and uterus [10]. MTNR1B consists of 363 amino acids and shares 60% homology with MTNR1A, and it is widely distributed in the mammalian brain, skin, heart, liver, kidney, ovaries, and testes [11, 12]. Over the past years, a significant number of studies have tended to focus on the effects of MTNR1A and MTNR1B on the regulation of reproduction in female animals, while ignoring the importance of male animals in the reproductive process.

In order to maintain the balance of hormone secretion in the reproductive system, the hypothalamus-pituitary-gonadal axis (HPGA) [13], an automatic feedback regulation system, functions in the hypothalamus, pituitary, and gonadal organs. This is known as the hypothalamus-pituitary-ovary axis (HPOA) in female animals and the hypothalamus-pituitary-testis axis (HPTA) in male animals. As a hormone related to animal reproduction, MT is generally considered an important factor in the regulation of seasonal reproductive activity and reproductive capacity in animals [14]. A variety of organs have been shown to synthesize and secrete small amounts of MT, including the testis and epididymis [15, 16], but the role of MT secreted by these tissues in male reproductive regulation remains unclear. Other studies have shown that the MTNR mediates the regulation of MT's effect on mammalian seasonal estrus, and the HPTA plays an important regulatory role in this process [17–19]. Therefore, clarifying the histomorphological characteristics of the HPTA is essential for elucidating the mechanism of reproductive regulation in male animals. In this study, we investigated the distribution and expression of MTNR1A and MTNR1B in the HPTA and their mechanisms of action in order to elucidate the role of MT in reproductive regulation in Tibetan sheep in plateau pastoral areas, thus providing data to support the development of effective methods to improve sheep reproductive performance.

Animals in highland pastoral areas live in short-day, low-oxygen environments year round, and their reproductive physiological characteristics differ greatly from animals raised in plain areas. Tibetan sheep (*Ovis aries*) live year-round on the Tibetan plateau at high altitudes, above 3000 m. These animals have germplasm characteristics such as tolerance to extreme cold and low oxygen as well as genetic stability, making them a valuable livestock resource in China [20]. Tibetan sheep are also typical seasonal estrus animals, which greatly impacts the development of agriculture and animal husbandry in the Tibetan areas of China due to their low reproductive rate. Small tail Han sheep (*Ovis. aries*) have also been introduced and bred in plateau pastoral areas. These animals exhibit both strong reproductive performance and estrus in four seasons [21]. The expression patterns of the MTNR1A and MTNR1B genes in the HPTA of seasonal estrus sheep and perennial estrus sheep have not been elucidated. Therefore, this study compared the expression patterns of two membrane receptor genes and reproductive hormone receptor (AR and ER) genes in testis and epididymis tissues of two estrous patterns

of sheep to clarify the relationship between MTNR1A and MTNR1B and seasonal reproduction of Tibetan sheep. On this basis, we hypothesized that the differential expression of MTNR1A and MTNR1B genes in different breeding seasons is an important factor affecting the seasonal estrus of male Tibetan sheep. The objective of this study was to investigate the specific role of MTNR1A and MTNR1B genes in seasonal reproduction of Tibetan sheep in plateau pastures.

## 2. Materials and methods

### 2.1 Experimental animals and sample collection

Samples were collected from the breeding cooperative in Datong District, Xining City, Qinghai Province. Healthy, well nourished and disease free adult rams were selected after health status examination. Tissues of fresh hypothalamus, hypophysis, testis, and epididymis (including the caput, corpus, and cauda) were obtained from adult male Tibetan sheep (2.5 years old, n = 8) and small tail Han Sheep (2.5 years old, n = 8) immediately after execution of rams by bloodletting through the carotid artery (methods of sacrifice) in the Datong District (Xining City, Qinghai Province, China). The breeding season samples were collected in August, and the non-breeding season samples were collected in February. A portion of each HPTA tissue sample was quickly frozen in liquid nitrogen, transported to the laboratory, and then stored at −80˚C for RNA and protein extraction, and portions of the remaining tissue were stored in 4% paraformaldehyde and 2.5% glutaraldehyde for histological and ultrastructural analyses, respectively. Blood (10 mL) was collected from the jugular vein of each animal using a heparin sodium anticoagulation vacuum blood collection tube (number of samples, > 10). Blood samples were centrifuged at 3500 rpm for 5 min, and the supernatant was transferred to a clean test tube, which was stored in a freezer at −20˚C for further analysis.

### 2.2 Drugs and reagents

All experimental antibodies were purchased from commercial suppliers. Sheep Melatonin ELISA kit (QS48694) was obtained from Qisong Biological, Beijing, China, and rabbit polyclonal antibodies anti-MTNR1A (bs-0027R) and anti-MTNR1B (bs-0963R) were purchased from Beijing BIOSS Antibodies Co., Ltd., China. Goat anti-rabbit IgG H&L (ab150077, Alexa Fluor® 488; ab150079, Alexa Fluor® 647; ab150080, Alexa Fluor® 594) were obtained from Abcam, Cambridge, UK. DAB color reagent kit (PA110) was obtained from Beijing TIANGEN Biotechnology Co., Ltd. The immunohistochemical staining kit (SP-0023) used was obtained from ZYMED USA, Beijing BIOSS Biotechnology Co., Ltd. ECL Plus ultrasensitive luminescent solution (PE0010) was purchased from Solebao Biotechnology Co., Ltd.

### 2.3 Immunohistochemistry and immunofluorescence

Immunohistochemical streptavidin-perosidase (SP) method: Epididymal tissues were embedded in paraffin and cut into 4-μm-thick sections. Paraffin sections were dewaxed and dehydrated, prepared with microwave oven antigen retrieval, blocked with 3% $H_2O_2$ solution for 10 min, and incubated with goat serum albumin for 15 min. Subsequently, 50 μL of rabbit polyclonal antibody (MTNR1A and MTNRIB) diluted 1:300 was added to each slide; the negative control consisted of 3% goat serum instead of the first antibody. The slides were incubated at 37˚C for 4 h, washed three times with phosphate-buffered saline (PBS) (each wash 5 min), and then 50 μL of biotin-labeled goat anti-rabbit IgG working solution was added, and the sections were incubated at 37˚C for 15 min and then washed three times with PBS (each wash 5 min). Horseradish peroxidase–labeled streptavidin solution was added, and the sections were

washed with PBS three times (each wash 5 min). DAB color developing solution was added for 5–20 min. Hematoxylin counterstaining was performed for 5 min, and then the sections were dehydrated using an alcohol gradient, made transparent with xylene, and sealed with neutral gum. Finally, the sections were observed under a microscope.

Double immunofluorescence: Immunofluorescence staining was performed with the primary antibody (MTNR1A); sections were incubated at 37˚C for 4 h and rinsed with PBS three times. Subsequent steps were completed in a dark room. Anti-Rabbit IgG H&L AF488 or AF594 (dilution ratio 1:1000) was added, incubated at 37˚C for 1 h, and the slides were then washed five times with PBS (each wash 5 min). The second antibody (MTNR1B) was then added, and the sections were incubated at 37˚C for 4 h and washed five times in PBS (each wash 5 min). Rabbit anti-PHD2/AF647 (dilution ratio 1:1000) was added dropwise and incubated at 37˚C for 1.5 h. After washing with PBS, DAPI was added dropwise, and the sections were incubated in a dark room for 10 min. After further washing with PBS, the patch was sealed with a capping agent, and the sections were observed under a laser confocal microscope. The negative control consisted of 0.01 mol/L PBS instead of the first antibody. The remaining conditions and steps were the same [22].

## 2.4 Electron microscopy sample preparation and observation

Scanning electron microscopy sample preparation: The testes and epididymis of Tibetan sheep and small tail Han sheep were cut into small pieces with length and width of 0.3 cm × 0.3 cm and 0.1 cm thickness and fixed with 2.5% glutaraldehyde for 2 days. The tissues were washed four times with 0.1 mol/L phosphate buffer (each wash 15 min). The samples were treated with 1% $OsO_4$ for 1 h and washed with double-distilled water six times (each wash 10 min). The tissues were then treated with 2% tannic acid for 30 min and washed with double-distilled water six times (each wash 10 min), subjected to gradient ethanol dehydration (30%, 50%, 70%, 80%, 90%, 95%, and 100%; each stage 30 min), and then soaked in isoamyl acetate for 30 min. Tissues were dried by critical point drying, and then the samples were sprayed with gold and observed using a scanning electron microscope.

Transmission electron microscopy sample preparation: The testes and epididymis of Tibetan sheep and small tail Han sheep fixed in 2.5% glutaraldehyde were cut into small pieces (0.2 cm × 0.2 cm × 0.2 cm) and fixed in 2% $OsO_4$ at 4˚C for 3 h. The pieces were dehydrated with a gradient acetone series (30%, 50%, 70%, 80%, 90%, 95%, and 100%) and then embedded in epoxy resin. Ultrathin sections were prepared and affixed to the copper mesh, stained with uranium acetate and lead citrate, and then observed and photographed using a JEM-100CX electron microscope (Japan NEC).

## 2.5 Determination of melatonin concentration

Pre-prepared serum samples were placed into the wells of a 96-well ELISA plate (150 μL per well) according to the instructions of the ELISA kit (Qisong Bio, Beijing, China). The OD of each well was measured using a microplate reader. A linear regression line for the standards was generated, and the concentration correlation coefficients (R values) were determined. The R values were >0.990. Melatonin concentrations are expressed as the mean ± standard deviation (M ± SD).

## 2.6 qRT-PCR analysis

Total RNA was extracted from HPTA tissues of Tibetan sheep and small tail Han sheep using an RNA extraction kit according to the manufacturer's instructions (ER501-01, Beijing Trans-Gen Biotech Co., Ltd., Beijing, China), and RNA integrity was assessed by agarose gel

**Table 1. List of the primers information used in qRT-PCR.**

| Gene | Sequences(5'→3') | Product length(bp) | Tm | Reaction efficiency | Accession no. |
|---|---|---|---|---|---|
| MTNR1A | F:GTGGTGGTGTTCCATTTCATAGT | 122 | 60°C | 97% | NM_001009725.1 |
|  | R:GGCTTTAGTTTCGGTTTGTTGT |  |  |  |  |
| MTNR1B | F:AGGTCAAGGCGGAGAGCAA | 148 | 58°C | 103% | NM_001130938.1 |
|  | R:GCCACTTCTTCGGGGTCAA |  |  |  |  |
| AANAT | F:CCTTCATCTCTGTCTCCGGC | 167 | 59°C | 104% | NM_001009461.1 |
|  | R:TGCCAGCGACTCCTGAGTAA |  |  |  |  |
| HIOMT | F:GCTCTTTATGCTCAGAAGGACTCAA | 107 | 58°C | 104% | NM_001306120.1 |
|  | R:ACAAGCTGATGGAACAGAGAACTG |  |  |  |  |
| AR | F: TTCAGTGGAAGAACCGAGCG | 152 | 54°C | 96% | NM_001308584.1 |
|  | R: TGTACGCAAACCTCTGGTGG |  |  |  |  |
| ERα | F:CGGTGGATGTGGTCCTTCTCTCT | 234 | 56°C | 94% | AY033393.1 |
|  | R:AGGGAAGCTCCTATTTGCTCC |  |  |  |  |
| β-actin | F:CCTCCAGCCTTCCTTCCTGG | 188 | 60°C | 102% | U39357.2 |
|  | R:GCCAGGGCAGTGACTTCTTT |  |  |  |  |

electrophoresis. cDNAs were synthesized using a reverse transcription kit (AT311-02, Beijing TransGen Biotech Co., Ltd., Beijing, China) and stored at −80°C. Primers were designed using primer design software (Primer Premier 5.0; primer Biosoft International, Palo Alto, CA, USA), and primer information is shown in Table 1. The primer sequences were derived from NCBI (www.ncbi.nlm.nih.gov). The cDNAs were diluted to the same concentration (300 ng/mL), and the upstream and downstream primers were diluted to 10 μmol/L. qRT-PCR was conducted using a Light Cycler480 thermocycler (Roche, Berlin, Germany). Using the sheep β-actin gene as an internal reference, the total reaction system was 20 μL, including 1 μL each of upstream and downstream primers, 1 μL of template (cDNA), 10 μL of polymerase, and 7 μL of ddH$_2$O. Reaction conditions: 95°C, 30 s, followed by 95°C, 5 s; 50–60°C, 15 s; and 72°C, 10 s. A total of 45 cycles were performed. At least three replicates were set for each sample to ensure the accuracy of the results.

## 2.7 Western blot analysis

Hypothalamus, pituitary gland, testis, and epididymis tissues of Tibetan sheep and small tail Han sheep stored at −80°C were taken out and ground into a fine powder using a freezing grinder. A total of 1 mL of tissue lysis solution, 1 μL of protease inhibitor, and 1 μL of phenyl-methanesulfonyl fluoride were added, and the tissues were repeatedly shaken five times for 10 min each in an ice box at low temperature. Samples were centrifuged at 4°C for 15 min using a low-temperature centrifuge, and the supernatant was removed and frozen at −80°C before use. A BCA protein quantification kit (PC0020, Solarbio Biotechnology Co., Ltd., Beijing, China) was used to quantify the extracted protein, and then each sample was diluted to the same concentration. Protein denaturation, electrophoresis, and membrane transfer were then performed. Following transfer, the PVDF membrane was removed and blocked with skimmed milk powder, after which the primary antibody was added (dilution ratio 1:400) and incubated at 37°C for 2 h. The membrane was then washed with Tris-buffered saline + Tween 20 (TBST) five times for 10 min each. The secondary antibody (dilution ratio of 1:3000) was then added and incubated at 37°C for 1.5 h, after which the membrane was washed with TBST five times for 10 min each. ECL chemiluminescent solution was then added dropwise to the membrane and developed under a chemiluminescence instrument.

## 2.8 Data statistics

Western blot data were quantified using ImageJ software (National Institutes of Health, Bethesda, Maryland, USA). qRT-PCR data were analyzed using the $2^{-\Delta\Delta CT}$ method, and the data were tested for significance using SPSS 17.0 statistical software (IBM Technology, Chicago, IL, USA). All data were found to be normally distributed by plotting histograms, and post hoc tests were performed using the "t" test. Histograms were plotted using Graphpad 9.0 software. The correlation heat map was analyzed and plotted using Origin 2021 software.

# 3. Results

## 3.1 Morphological observation of HPTA in Tibetan sheep

The morphological structure of HPTA tissue in Tibetan sheep was observed by H & E staining. The results showed that the hypothalamus of Tibetan sheep was dominated by glial cells with different shapes (Fig 1A). The adenohypophysis was dominated by chromophobe cells, with a small number of eosinophilic and basophilic cells that together formed multiple cell clusters distributed in the adenohypophysis (Fig 1B). Many spermatogonia and primary spermatocytes were observed in the testis of Tibetan sheep. The seminiferous tubule was surrounded by 3–4 layers of myoid cells and closely connected with Leydig cells (Fig 1D). The epididymis of Tibetan sheep primarily consisted of the epididymal duct and epididymal mesenchyme. The principal cells were closely arranged into a tube, which is the basic skeleton of the epithelium of the epididymal duct, whereas the basal cells and halo cells were evenly distributed outside the tube formed by the principal cells, and a large number of cilia were distributed inside the tube (Fig 1D–1F).

Scanning electron microscopy showed that cells were pancake-shaped and uniformly distributed in Tibetan sheep hypothalamic tissue (Fig 2A). The chromophobe cells in pituitary tissue were also pancake-shaped. Moreover, a large number of filamentous nerve fibers were

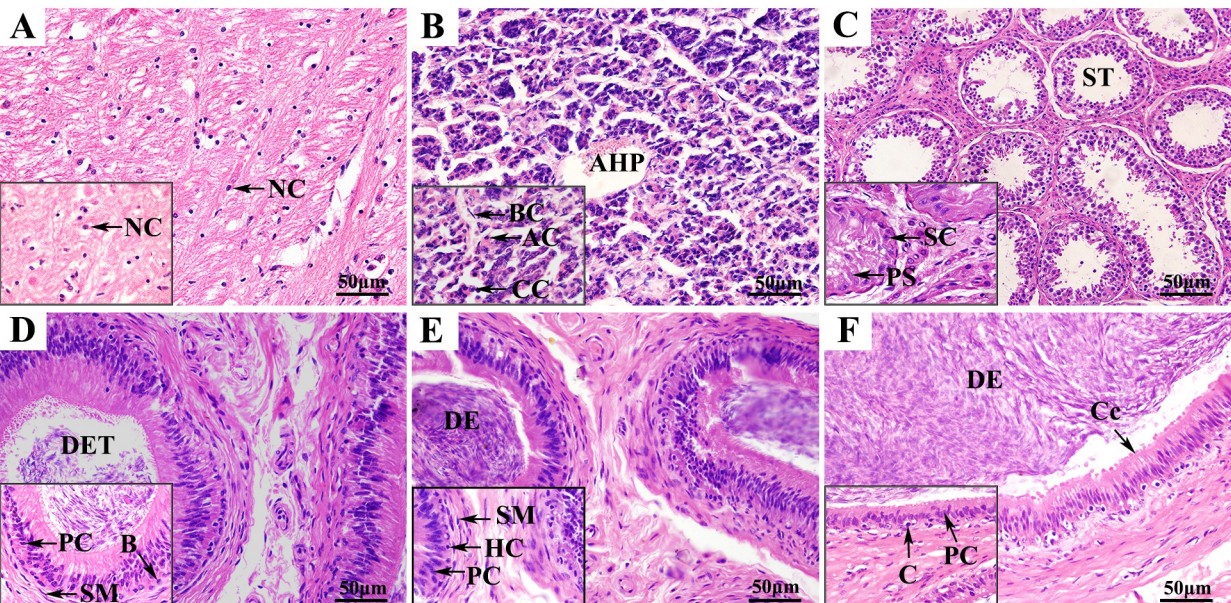

**Fig 1. H & E staining of Tibetan sheep HPTA tissues.** (A) Hypothalamus, (B) pituitary, (C) testis, (D) epididymis head, (E) epididymis body, (F) epididymis tail. NC, neurogliocyte; Cp, capillary; AC, acidophilic cell; BC, basophilic cell; CC, chromophobe cell; FS: follicle; NF, nerve fiber; ST, seminiferous tubule; MC, myoid cell; LC, Leydig cell; SC, spermatogonium; PS, primary spermatocyte; C, cilium; S, spermatozoon; DET, ductuli efferentes testis; B, basal cell; PC, principal cell; SM, smooth muscle cell; DE, ductus epididymidis.

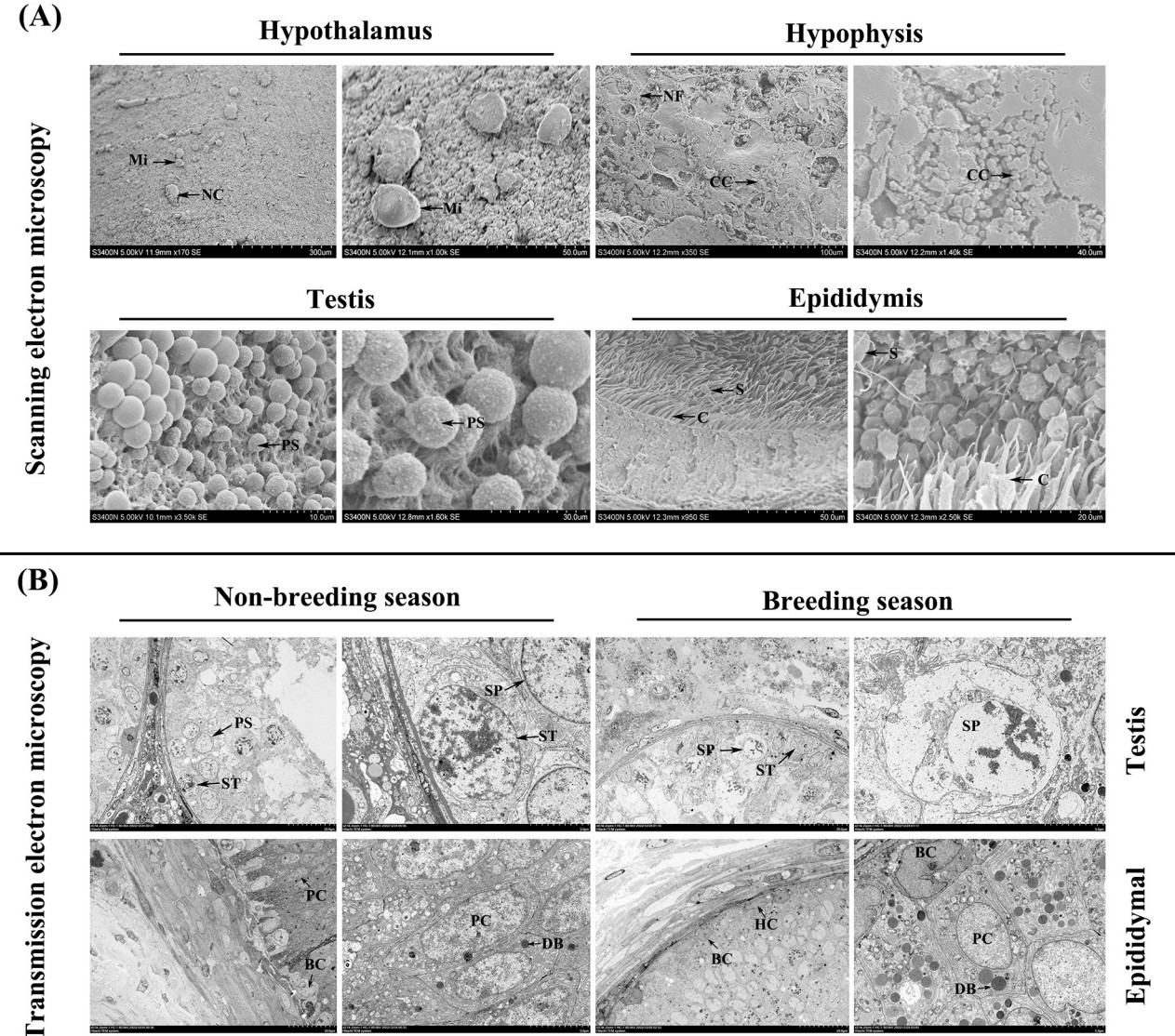

**Fig 2. Electron microscopy images of Tibetan sheep HPTA.** (A) Scanning electron microscopy images of HPTA; (B) Transmission electron microscopy ultrastructure images of the testis and epididymis of Tibetan sheep in different breeding seasons. NF, nerve fiber; CC, chromophobe cell; SC, spermatogonium; PS, primary spermatocyte; ST, spermatozoon; C, cilium; Sc, Sertoli cell; PC, principal cell; BC, basal cell; HC, halo cell; DB, dense body.

distributed in the adenohypophysis (Fig 2A). The primary spermatocytes in the Tibetan sheep testis were spherically attached to the wall of the seminiferous tubules (Fig 2A). The Tibetan sheep epididymis was composed of epithelial tissue, connective tissue, and epididymal mesenchyme, and the epididymal ducts contained a large number of mature spermatozoa (Fig 2A). Cilia were clearly visible inside the epididymal duct (Fig 2A).

## 3.2 Ultrastructural changes in the testis and epididymis of Tibetan sheep in different breeding seasons

Transmission electron microscopy showed that spermatogonia were more active in the testes of Tibetan sheep during the breeding season than during the non-breeding season, and the

number of spermatocytes in the meiotic phase was significantly increased in the testis during the breeding season compared to the non-breeding season (Fig 2B). There was a significant thickening of connective tissue in the epididymis of Tibetan sheep during the breeding season, and the dense bodies of the epithelial principal cells of the epididymis were also more abundant than in the non-breeding season, with no significant differences in the ultrastructure of other cells (Fig 2B).

### 3.3 MT concentration in the blood of sheep in different breeding seasons

The mean blood MT concentration was determined by ELISA, which showed that the mean blood MT concentration in Tibetan sheep was significantly higher during the breeding season than in the non-breeding season and significantly higher than the concentration in the blood of small tail Han sheep ($p<0.01$). However, the differences in MT concentrations in the blood of small tail Han sheep during the breeding season were not significant ($p>0.05$, Table 2).

### 3.4 Analysis of the localization and expression of MTNR1A and MTNR1B in the sheep HPTA

The results of immunohistochemical analyses showed that MTNR1A and MTNR1B were expressed in all tissues of the Tibetan sheep HPTA. In the hypothalamus, MTNR1A and MTNR1B were predominantly distributed in the cytoplasm of oval cells. (Fig 3A1 and 3A2). In addition, MTNR1A was strongly expressed around hypothalamic capillaries (Fig 3A1). In the adenohypophysis, MTNR1A and MTNR1B were distributed in the cytoplasm of chromophobe cells and around the nucleus of eosinophils and strongly positively expressed in the cytoplasm of chromophobe cells (Fig 3B1 and 3B2). In the testis, MTNR1A and MTNR1B were distributed in the seminiferous tubules. MTNR1A was strongly positively expressed around the nucleus of primary spermatocytes and weakly positively expressed in the testicular interstitium (Fig 3C1), whereas MTNR1B was strongly positively expressed in spermatogonia, primary spermatocytes, spermatozoa, supporting cells, and the testicular interstitium (Fig 3C2). MTNR1A and MTNR1B were also distributed in the caput, corpus, and cauda of the Tibetan sheep epididymis, whereas MTNR1A was mainly distributed around epididymal duct cilia, capillary endothelial cells, basal cells, and halo cells (Fig 3D1, 3E1 and 3F1). MTNR1B was widely distributed in principal cells and interstitial cells (Fig 3D2, 3E2 and 3F2) in addition to cilia, basal cells, and halo cells of the epididymal duct.

Immunofluorescence analyses showed strong positive expression of MTNR1A and MTNR1B in all tissues of the Tibetan sheep HPTA. The fluorescence intensity of MTNR1A was significantly higher than that of MTNR1B in the pituitary, testis, epididymis caput, and epididymis corpus, but the opposite was observed in the hypothalamus and epididymis cauda. Moreover, the immunofluorescence of MTNR1A was more specific than that of MTNR1B in all tissues of the HPTA. Particularly in the epididymis, MTNR1A was specifically expressed in

**Table 2. Melatonin levels in sheep blood in different breeding seasons (M±SD).**

| Factor | Group | Breeding season | Non-breeding season |
|---|---|---|---|
| Average MT level (pg/mL) | Tibetan sheep | 13.819±0.554[Aa] | 10.223±0.216[Bb] |
| | Small Tail Han sheep | 9.243±0.439[Bc] | 8.958±0.762[Bc] |

Note: The number of samples per group was >10, and each sample was tested three times. M: mean; SD: standard deviation. Different lowercase letters show significant differences ($p<0.05$); different uppercase letters indicate significant difference ($p<0.01$).

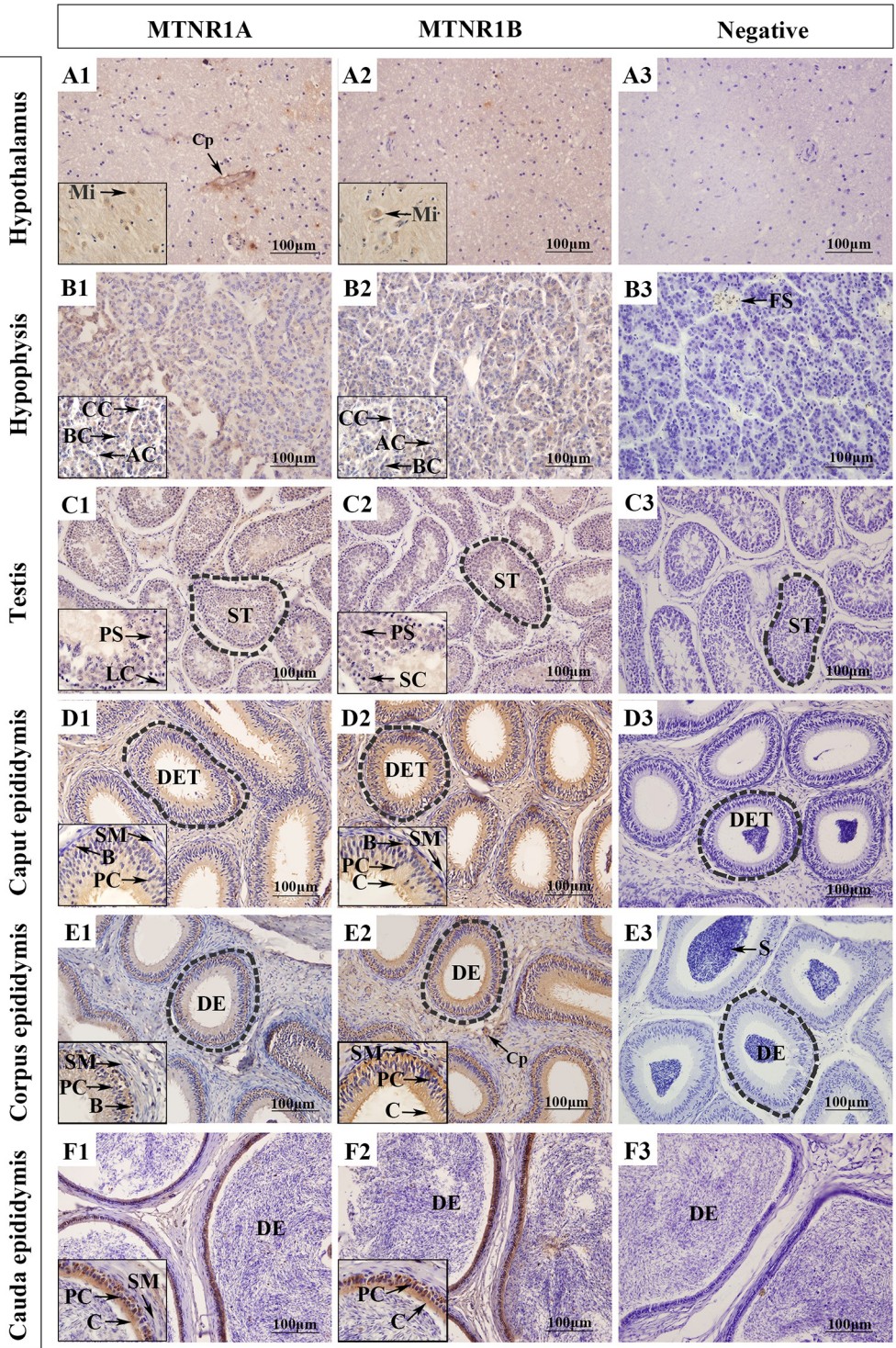

**Fig 3. Tissue localization of MTNR1A and MTNR1B proteins in various tissues of the HPTA of Tibetan sheep.**
(A1-F1) Tissue localization of MTNR1A protein in ram HPTA. (A2-F2) Tissue localization of MTNR1B protein in ram HPTA. The magnification of the main images is 200×. The boxes in the lower left corners are partial enlarged views (1000×). Cp, capillary; AC, acidophilic cell; BC, basophilic cell; CC, chromophobe cell; FS: follicle; NF, nerve fiber; ST, seminiferous tubule; MC, myoid cell; LC, Leydig cell; SC, spermatogonium; PS, primary spermatocyte; C, cilium; S, spermatozoon; DET, ductuli efferentes testis; B, basal cell; PC, principal cell; SM, smooth muscle cell; DE, ductus epididymidis.

the cytoplasm of basal and halo cells of the epididymal epithelium (Fig 4). Immunofluorescence results also showed that MTNR1A and MTNR1B co-localized in the cytoplasm of epididymal basal and halo cells, and the immunopositive cells formed a natural closed cell barrier around the epididymal duct (Fig 4).

## 3.5 Expression levels of MTNR1A mRNA and MTNR1B mRNA and protein in HPGA tissues from the different breeds of sheep

The expression levels of *MTNR1A* mRNA and *MTNR1B* mRNA and protein in HPGA tissues of the two types of sheep were analyzed by qRT-PCR and western blotting. The results showed that the expression of *MTNR1A* mRNA and *MTNR1B* mRNA and protein differed significantly in HPGA tissues of the different sheep breeds ($p < 0.05$, Fig 5). *MTNR1A* mRNA was expressed at the highest level in the adenohypophysis of Tibetan sheep, followed by the testis, epididymis cauda, epididymis corpus, epididymis caput, and hypothalamus tissue (Fig 5A), whereas *MTNR1B* mRNA was expressed at the highest level in Tibetan sheep testis, followed by the adenohypophysis, epididymal cauda, epididymal corpus, epididymal caput, and hypothalamus. However, MTNR1A protein expression was not different in the pituitary, testis, epididymal caput, and epididymal cauda of Tibetan sheep, but all were significantly higher compared with the hypothalamus. Furthermore, the expression of MTNR1B protein was significantly higher in the pituitary and cauda epididymis of Tibetan sheep than in other tissues (Fig 5A).

In the HPTA of small tail Han sheep, *MTNR1A* and *MTNR1B* mRNAs were expressed at the highest levels in the pituitary and testis, followed by the epididymis cauda, epididymal caput, epididymal corpus, and hypothalamus tissue (Fig 5B). However, MTNR1A protein expression and *MTNR1A* mRNA expression showed opposite trends in the epididymal caput and testis, with consistent expression in other tissues (Fig 5B). Similarly, MTNR1B protein and mRNA expression showed opposite trends in the testis and epididymal caput, with consistent expression in other tissues (Fig 5B).

## 3.6 Differential expression of reproductive hormone receptor genes in sheep testis and epididymis in different reproductive stages

The expression levels of genes encoding the MT receptor, MT receptor synthase, and reproductive hormones were measured by qRT-PCR in the testis and epididymis of two different breeds of sheep in different breeding seasons. qRT-PCR product size was determined by agarose gel electrophoresis, and the results showed that the product matched the size of the target gene fragment (Fig 6). qPCR results showed that *MTNR1A* and *AANAT* mRNA expression was significantly upregulated in both the testis and epididymis of Tibetan sheep during the breeding season ($p < 0.01$, Fig 7A), whereas *MTNR1B* and *HIOMT* mRNA expression was upregulated in the testis, epididymal caput, and epididymal cauda but downregulated in the epididymal corpus ($p < 0.01$, Fig 7A). In Tibetan sheep, *AR* mRNA expression was significantly upregulated in the testis, epididymal caput, and epididymal cauda during the breeding season, and although it was also upregulated in the epididymal corpus, the difference was not significant. *ERα* mRNA expression was downregulated in the testis and epididymal corpus and upregulated in the epididymal caput and epididymal cauda during the breeding season in Tibetan sheep; although expression was not significantly upregulated in the epididymal corpus, it was significantly upregulated in the epididymal cauda ($p < 0.01$, Fig 7A). *MTNR1A* mRNA expression did not differ significantly between the testis and epididymal caput of small tail sheep during the breeding and non-breeding seasons, but there was a downregulation trend in epididymal bodies and upregulation in the epididymal tail during the breeding season

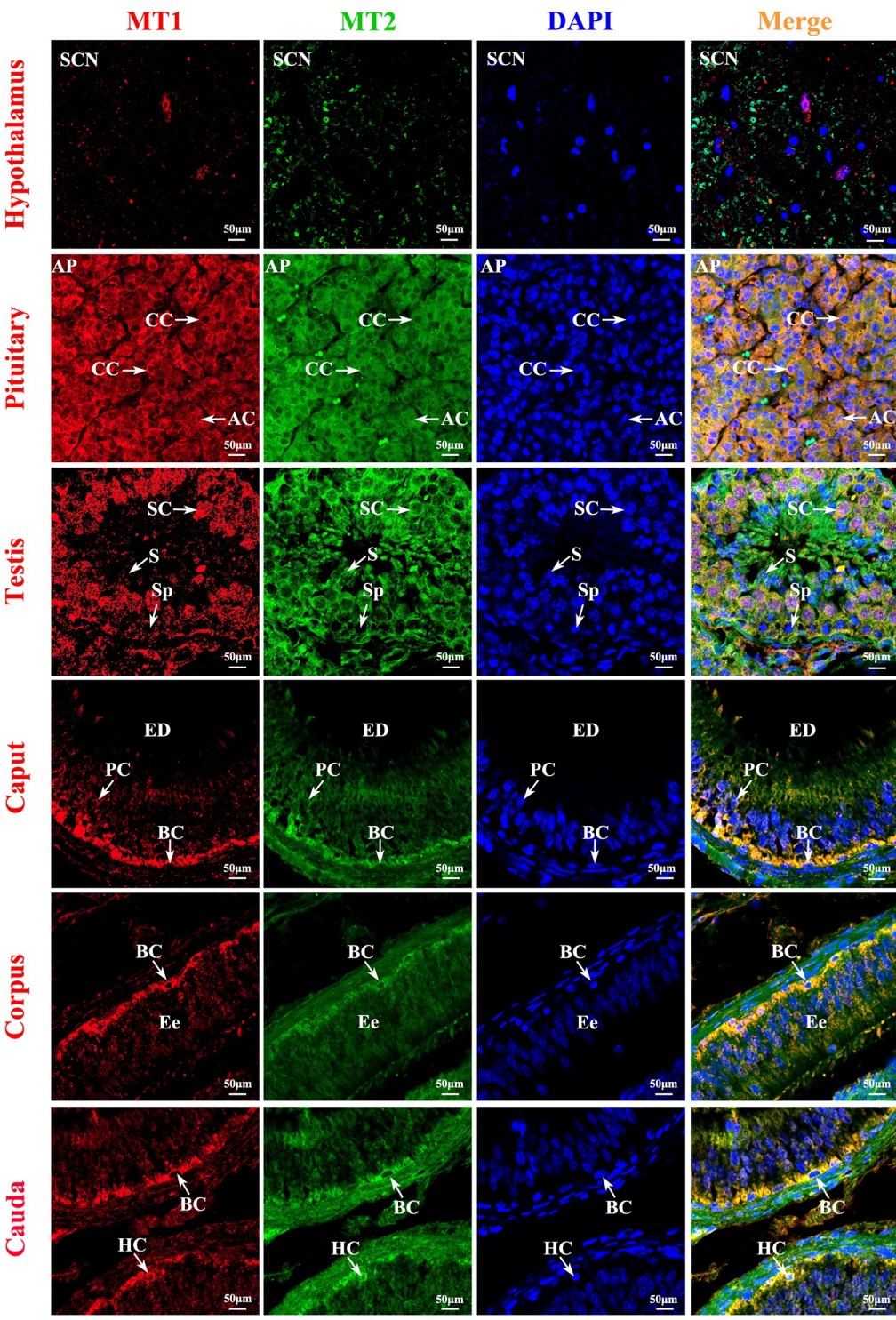

**Fig 4. Immunofluorescence co-localization of MTNR1A and MTNR1B proteins in various tissues of the HPTA of Tibetan sheep (400×).** The bar graphs on the right side of images show the protein immunofluorescence expression intensity. SCN, suprachiasmatic nucleus; CC, chromophobe cell; AC, acidophilic granulocyte; SC, Sertoli cells; S, spermatid; Sp, spermatogonium; PS, primary spermatocyte; PC, principal cell; BC, basal cell; HC, halo cell. **$p < 0.01$; *$p < 0.05$; ns (not significant), $p > 0.05$.

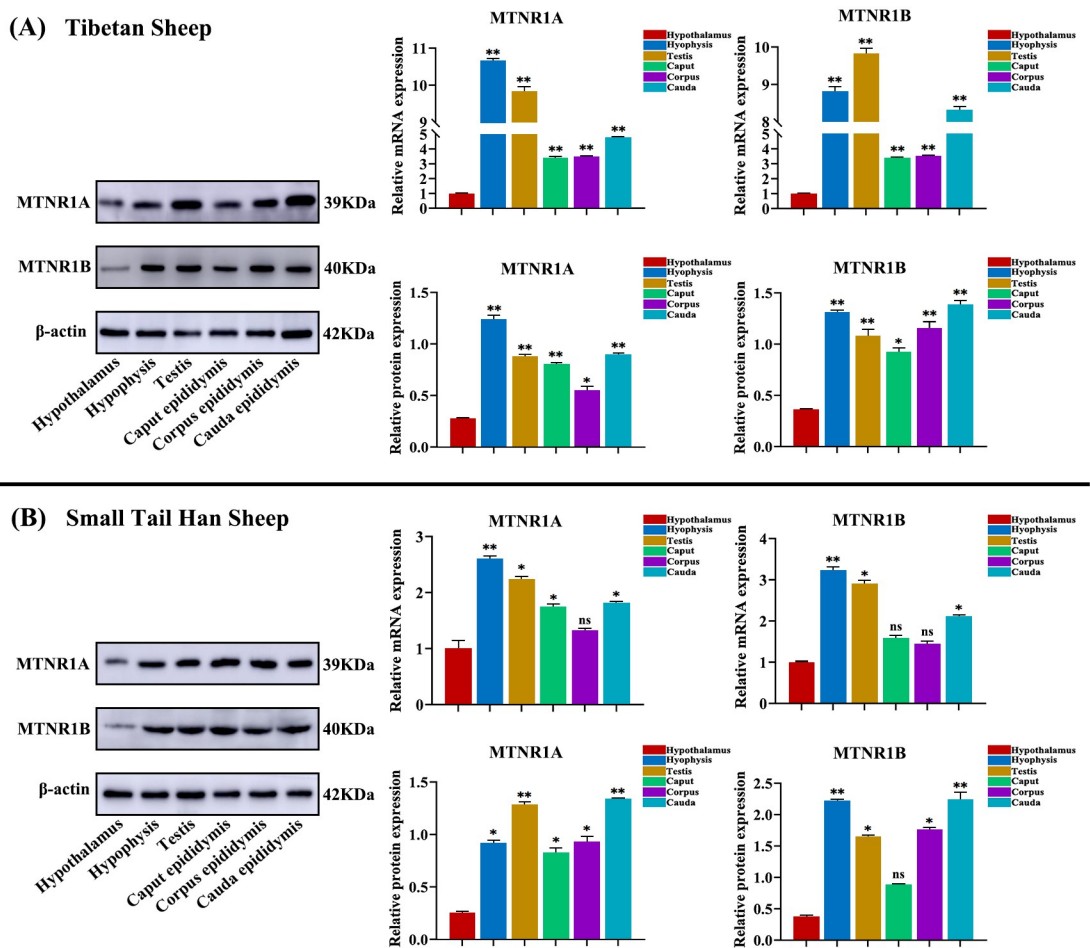

**Fig 5. Expression levels of *MTNR1A* mRNA and *MTNR1B* mRNA and protein in the HPGA tissues of adult Tibetan sheep and small tail Han sheep.** (A) Expression of *MTNR1A* mRNA and *MTNR1B* mRNA in HPGA tissues of Tibetan sheep. (B) Expression of *MTNR1A* mRNA and *MTNR1B* mRNA in HPGA tissues of small tail Han sheep. Hypothalamic *MTNR1A* mRNA and *MTNR1B* mRNA and protein expression levels were examined as controls. The expression of β-actin was used as an endogenous control. **$p< 0.01$; *$p<0.05$; ns (not significant), $p>0.05$.

($p<0.01$, Fig 7B). *MTNR1B* mRNA expression was significantly upregulated in the testis, epididymal caput, and epididymal corpus of the small tail Han sheep during the breeding season ($p<0.01$, Fig 7B). *AANAT* and *HIOMT* mRNA expression was not significantly upregulated in the testis or epididymal corpus of the sheep during the breeding season, but *HIOMT* mRNA expression was significantly upregulated in the epididymal tail during the breeding season ($p<0.01$, Fig 7B). In addition, *AR* mRNA expression was upregulated in the testis and epididymis of small tail Han sheep during the breeding season, but the difference was not significant. In contrast, the upregulation of *ERα* mRNA expression was significant ($p<0.01$, Fig 7B).

## 3.7 Analysis of the correlation in expression of reproductive hormone receptor genes in the sheep testis and epididymis

Analyses using Origin software revealed that the expression of *MTNR1A* mRNA in the testis and epididymis of Tibetan sheep in different breeding seasons was significantly positively correlated with the expression of *AANAT*, *HIOMT*, and *AR* mRNAs and negatively correlated

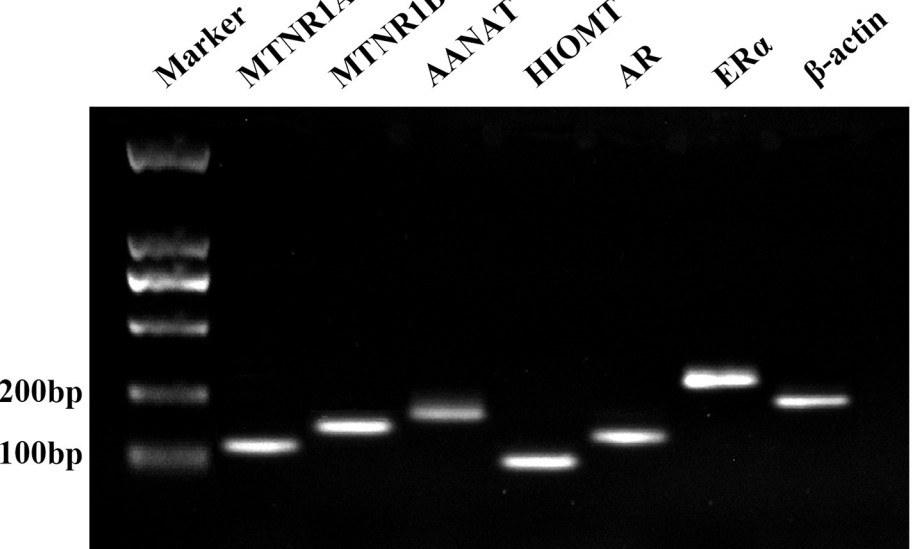

**Fig 6. Agarose gel electrophoresis analysis of qRT-PCR products.**

with the expression of *ERα* mRNA. Although *MTNR1B* mRNA expression was significantly and positively correlated with *AANAT*, *HIOMT*, and *AR* mRNA expression in the testis and epididymis cauda, the correlation was not significant in the caput and corpus of the epididymis (Fig 8A–8D). In addition, the expression of *MTNR1A* and *MTNR1B* mRNAs in the HPTA of small tail sheep was positively correlated with the expression of several other proteins in the epididymal caput, whereas either no correlation or an insignificant correlation was found for all other tissues (Fig 8E–8H).

## 4. Discussion

The HPTA plays an important role in the synthesis and secretion of reproduction-related hormones in male animals [23]. Moreover, it regulates the reproductive activity of animals via hormones [24]. Therefore, a thorough understanding of HPTA tissue characteristics is necessary to elucidate reproductive regulatory mechanisms. The organization of the HPTA varies according to species and environmental characteristics. Different animal species exhibit environment-specific changes in HPTA histochemical characteristics, and the cytoarchitecture of the HPTA is affected by a variety of environmental factors [25–27]. In this study, scanning electron microscopy analyses indicated that the division phase of spermatocytes is markedly increased in the testis of Tibetan sheep during the breeding season compared with the non-breeding season. In addition, we observed a significant thickening of connective tissue and an abundance of principal cell–dense bodies in the epididymis during the breeding season, indicating that the structural characteristics of the HPTA in Tibetan sheep also undergo adaptive seasonal changes. Consequently, studies of HPTA tissue characteristics and seasonal changes could provide a foundation for studies of the reproductive regulation mechanism.

Our previous study found that the effects of MT are mediated by MTNR1A, which acts on the HPOA to regulate reproductive activity in female Tibetan sheep [28]. However, little is known about the effects of MTNR1A and MTNR1B on the HPTA in Tibetan sheep. In this study, we found that both MTNR1A and MTNR1B are distributed in the HPTA of Tibetan sheep at particularly high levels in the pituitary gland and testis, and the distribution in the

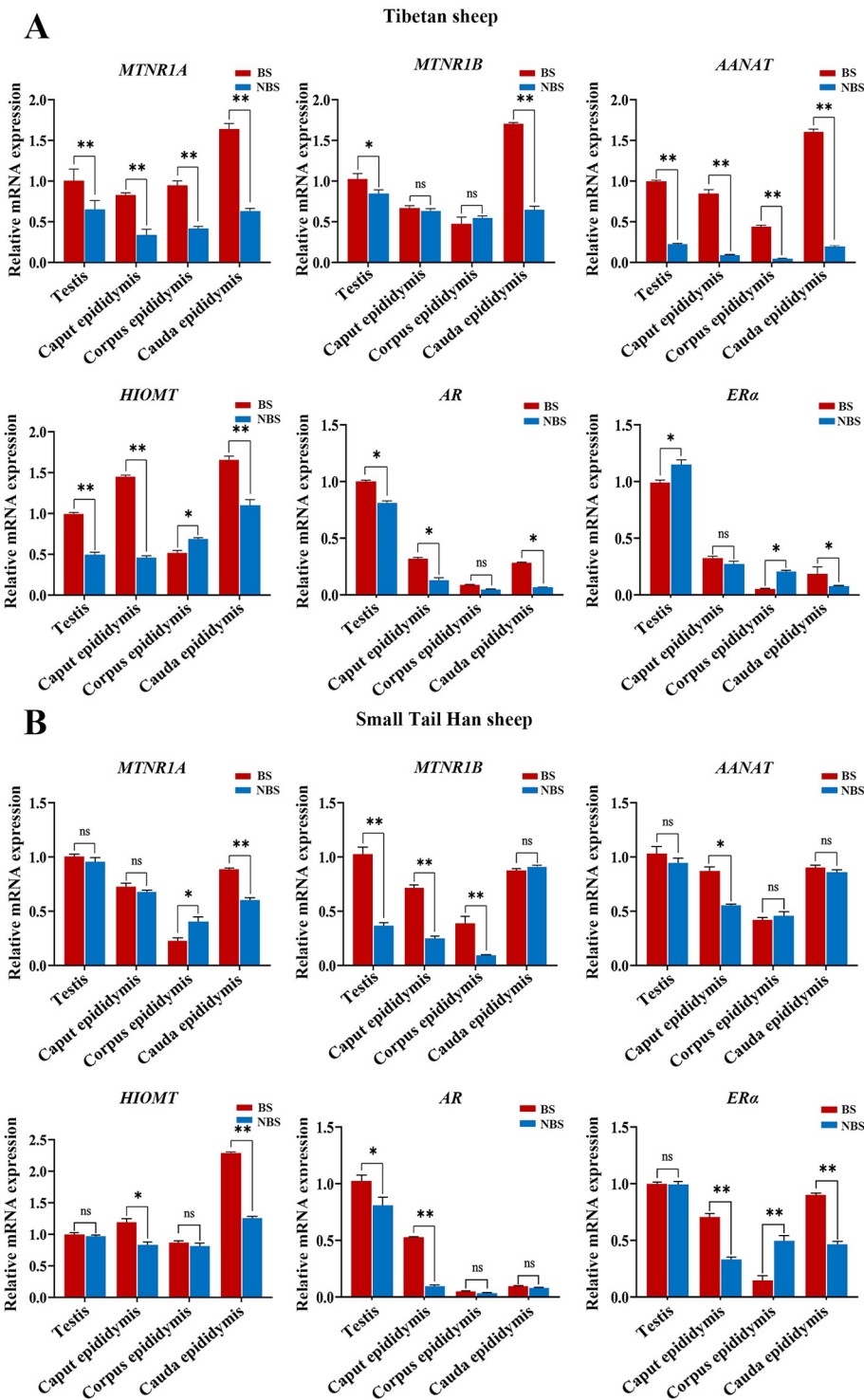

**Fig 7. Expression levels of MT receptor and reproductive hormone receptor genes in the testis and epididymis of sheep in different breeding seasons.** (A) Expression levels of MT receptor and reproductive hormone receptor genes in the testis and epididymis of Tibetan sheep. (B) Expression levels of MT receptor and reproductive hormone receptor genes in the testis and epididymis of small tail Han sheep. Testicular tissue in the breeding season was used as the control, and the β-actin gene was used as an internal reference gene. BS: breeding season; NBS: non-breeding season. **$p < 0.01$; *$p < 0.05$; ns (not significant), $p > 0.05$.

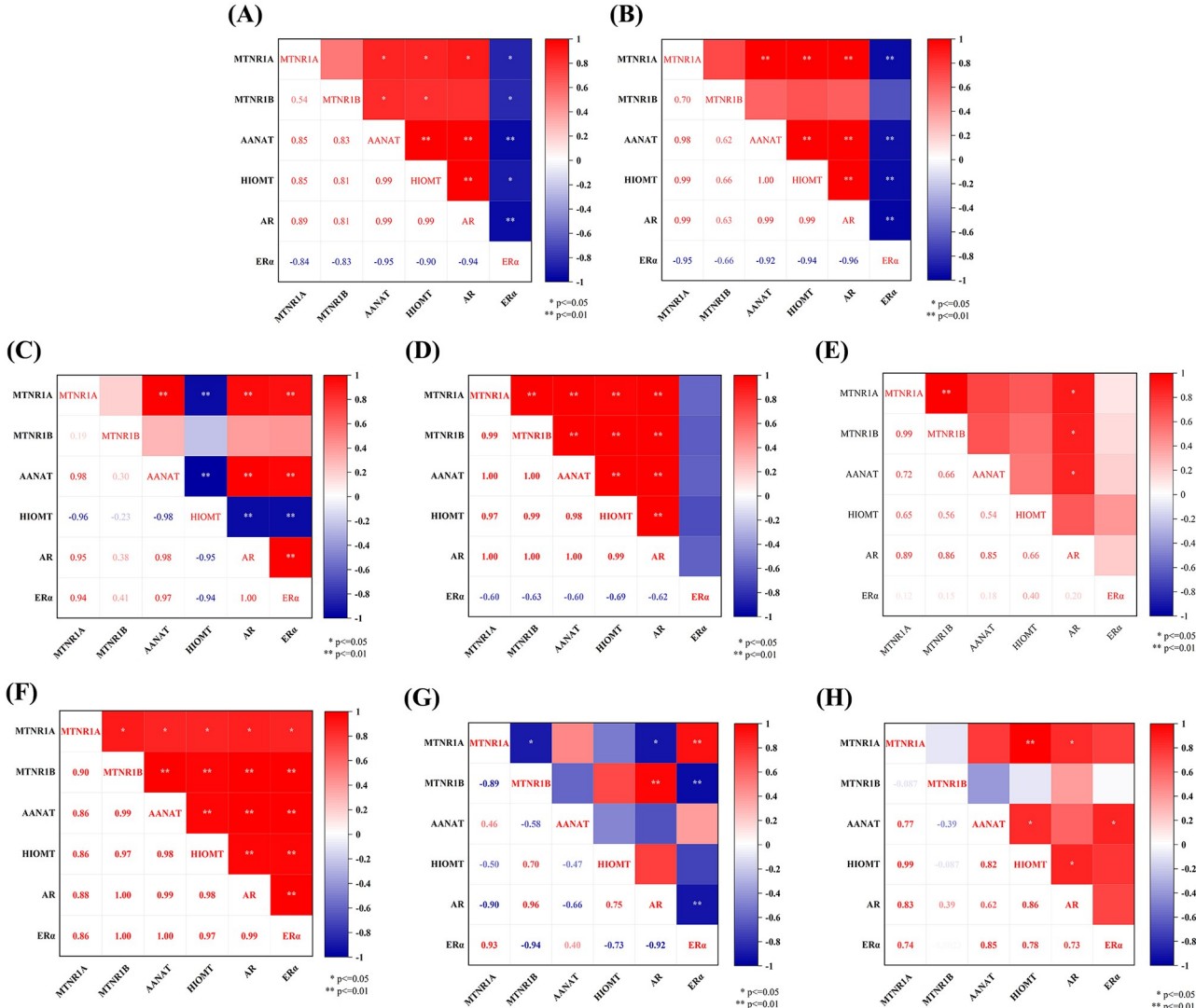

**Fig 8. Heat map of gene expression correlation analysis results.** A, B, C, and D represent Tibetan sheep testis, epididymal caput, epididymal corpus, and epididymal cauda, respectively. E, F, G, and H represent the correlation analysis of mRNAs in the small tail Han sheep testis, epididymal caput, epididymal corpus, and epididymal cauda, respectively.

epididymis is specific. MTNR1A and MTNR1B thus appear to play a crucial role in the HPTA of Tibetan sheep.

Many genes in the hypothalamus are involved in the photoperiodic regulation of reproductive activity in sheep [29]. Studies have shown that regulation of the effect of MT on the reproductive function of sheep is mediated by the premammillary hypothalamus [30]. In this study, we found that both MTNR1A and MNR1B are expressed in the hypothalamus tissue of Tibetan sheep. MTNR1A is expressed in the cytoplasm of hypothalamic oval cells, and we suspect that these oval cells are microglia. Studies have shown that microglia are abundantly distributed in the hypothalamus, and that microglia play an important role in the secretion of neurotrophic factors, the promotion of the development of the nervous system, and the promotion of neuronal survival and alteration of neuron numbers [31–34]. This suggests that MT

in the hypothalamus primarily originates from the blood circulation and that microglia may be involved in regulating the secretion of MT seasonally.

The reproductive activity of animals is mediated by reproductive hormones such as androgens and some steroid hormones such as estrogen [35, 36]. The pituitary gland controls the secretion of many hormones, such as gonadotropin hormone (GTH), thyroid-stimulating hormone, and growth hormone. Hypothalamic gonadotropin-releasing hormone (GnRH) secretion is promoted or inhibited by MT, and in turn, GnRH regulates the secretion of GTH by the adenohypophysis through the pituitary portal system. GTH includes luteinizing hormone and follicle-stimulating hormone, and in males, luteinizing hormone acts on testicular mesenchymal cells, whereas follicle-stimulating hormone acts on testicular supporting cells, in both cases promoting sperm production [9, 37]. In this study, the protein and mRNA expression levels of MTNR1A and MTNR1B were significantly higher in the pituitary gland than other tissues, indicating that the pituitary gland is an important target organ of MT. Studies have shown that the MT receptors MTNR1A and MTNR1B participate in the synthesis of testosterone in mice by regulating multiple cellular pathways [38, 39]. MTNR1A and MTNR1B are also highly expressed in human sperm and mediate the movement of human sperm cells [40]. In this study, MTNR1A and MTNR1B were found to be strongly expressed in the testis of both breeds of sheep, and MTNR1A was specifically expressed in testicular primary spermatocytes, whereas MTNR1B was strongly expressed in both primary and secondary spermatocytes as well as sperms, suggesting that MTNR1A mediates further division of primary spermatocytes in sheep, whereas MTNR1B may mediate sperm deformation and migration. MT regulates the development and function of Holstein bovine testis support cells by binding to MTNR1A and MTNR1B [41]; however, we did not find strong expression of MTNR1A and MTNR1B in sheep Sertoli cells, which may be related to species differences.

The epididymis is an important site for sperm maturation, processing, and storage [42, 43]. In this study, we found that MTNR1A and MTNR1B were strongly expressed in the epithelium, cilia, and pericapillary of the epididymis in sheep and co-expressed in the cytoplasm of epididymal epithelial basal cells and halo cells, particularly basal cells, suggesting that MTNR1A and MTNR1B play an important role in sperm maturation and processing. Basal cells play an important role in formation of the blood-epididymal barrier [44, 45], while halo cells are considered members of the epididymal immune cell family [46], and both cell types play important roles in formation of the blood-epididymal barrier and maintaining homeostasis of the epididymal immune microenvironment from various immune attacks [47]. In the present study, both scanning and transmission electron microscopy analyses showed that the epididymal basal cells were oval and arranged in the epithelial basement membrane. Fluorescence cytochemistry analyses showed that MTNR1A- and MTNR1B-immunopositive cells (basal cells and halo cells) formed a closed cellular barrier around the basement membrane of the epididymal duct, suggesting that MTNR1A and MTNR1B are involved in formation of the blood-epididymal barrier in sheep and play a crucial role in development of the immune microenvironment of the epididymis.

The multiple biological functions of MT are mediated by its membrane receptors [48]. For example, MTNR1A mediates MT-associated regulation of seasonal reproduction in ewes [49, 50]. Injection of exogenous MT during the non-breeding period increases plasma testosterone levels in rams [51, 52]. In view of the close relationship between MT and regulation of reproduction in sheep, the effect of MT on seasonal reproduction of Tibetan sheep in plateau pastoral areas was examined by measuring MT levels in the blood of male Tibetan sheep and small tail Han sheep in different seasons using ELISA. MT levels in the blood of Tibetan sheep exhibiting seasonal estrus were significantly higher in the breeding period (summer) than in the non-breeding period (winter) ($p < 0.01$), whereas MT levels in the blood of small tail Han

sheep exhibiting perennial estrus did not significantly differ between summer and winter ($p>0.05$). These data suggest that the seasonal reproductive activity of Tibetan sheep is regulated by internal MT levels. Transmission electron microscopy results indicated that more primary spermatocytes were at the division stage in Tibetan sheep during the breeding season compared with the non-breeding season, suggesting that the reproductive performance of Tibetan sheep is affected by seasonal factors. Studies have shown that testosterone levels, spermatogenesis, and fertility are also influenced by exogenous MT [53, 54]. Therefore, the regulation of seasonal reproductive activity by MT in sheep cannot be ignored. In the present study, qRT-PCR was used to measure the mRNA expression of *AANAT*, *HIOMT*, and *AR* and revealed that these genes are significantly upregulated along with upregulation of *MTNR1A* mRNA expression during the reproductive period in the testis and epididymis of Tibetan sheep. In contrast, these genes were not significantly upregulated in the testis and epididymis of small tail Han sheep, and *MTNR1B* mRNA expression was also not significantly upregulated. Consequently, our study clearly verified that the seasonal reproductive performance of Tibetan sheep is regulated by MT. In addition, significant positive correlations were found between the mRNA expression of *MTNR1A* and that of *AANAT*, *HIOMT*, and *AR* in Tibetan sheep in seasonal estrus, whereas the expression of *ERα* mRNA was significantly negatively correlated, indicating that MTNR1A is closely related to the function of reproductive hormones and also functions as a major factor in the regulation of seasonal reproductive activity in sheep.

A number of studies have demonstrated that MTNR1A gene polymorphisms are associated with seasonal estrus in female sheep [55–57] and may even affect litter size [58]. Similarly, *MTNR1A* gene polymorphisms affect the reproductive performance of rams [59]. In most cases, estrus in sheep is influenced by the estrus of ewes. Hence, we suggest that *MTNR1A* gene polymorphisms and the reproductive performance of seasonally breeding sheep are closely related. The results of this study provide data support for elucidating the mechanism of gonadal axis regulation in seasonally estrous animals, in addition to providing a theoretical basis for rationally utilizing seasonal breeding opportunities to improve the productivity of seasonally breeding economic animals.

## 5. Conclusion

MTNR1A and MTNR1B are the prominent membrane receptors mediating the regulatory role of MT in the HPTA of sheep. Seasonal reproductive activity in highland Tibetan sheep is regulated by high levels of MT. In addition, fluctuations in *MTNR1A* mRNA expression in the testis and epididymis of Tibetan sheep exhibit a significant positive correlation with seasonal expression of reproductive hormone receptors. Accordingly, MTNR1A and MTNR1B are important regulators of the HPTA in Tibetan sheep. Our results provide a reliable foundation for studies of the reproductive regulatory mechanism of seasonal animals in plateau pastoral areas.

## Supporting information

**S1 Raw images. Original images for gels.**
(ZIP)

## Acknowledgments

We thank Institute of Animal Husbandry and Veterinary Medicine, Chinese Academy of Agricultural Sciences for helping TEM measurements. We thank all the colleagues who assisted in

this study. We would like to thank The Charlesworth Group (www.cwauthors.com.cn) Editing Services for English language editing.

## Author Contributions

**Conceptualization:** Dapeng Yang, Ligang Yuan.

**Data curation:** Shaoyu Chen.

**Formal analysis:** Ligang Yuan.

**Investigation:** Xiaojie Ma.

**Methodology:** Dapeng Yang.

**Project administration:** Dapeng Yang, Ligang Yuan.

**Resources:** Guojuan Chen, Juanjuan Song.

**Supervision:** Shaoyu Chen.

**Validation:** Yindi Xing.

**Visualization:** Dapeng Yang.

**Writing – original draft:** Dapeng Yang.

**Writing – review & editing:** Ligang Yuan.

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
