## [Decision Letter · Decision Letter 0]

3 Jul 2023

PONE-D-23-06874Expression and Role of Melatonin Membrane Receptors in the Hypothalamic-Pituitary-Testicular Axis of Tibetan Sheep in a Plateau Pastoral AreaPLOS ONE

Dear Dr. Yang,

Thank you for submitting your manuscript to PLOS ONE. After careful consideration, we feel that it has merit but does not fully meet PLOS ONE’s publication criteria as it currently stands. Therefore, we invite you to submit a revised version of the manuscript that addresses the points raised during the review process.

In particular, it is important that you provide clear results about the morphological characteristics of the tissus and a complete description of the materials and methods used in the study. The other comments also provide constructive guidance for improving your manuscript.

We look forward to receiving your revised manuscript.

Kind regards,

Martine Migaud

Academic Editor

PLOS ONE

Journal Requirements:

3. To comply with PLOS ONE submissions requirements, in your Methods section, please provide additional information regarding the experiments involving animals and ensure you have included details on (1) methods of sacrifice, (2) methods of anesthesia and/or analgesia, and (3) efforts to alleviate suffering.

This study was supported by The Fund of the Discipline Team Project of Gansu Agricultural University, grant number: GAU-XKTD-2023; The Project of “Innovation Star” for Excellent Graduate Students in Gansu Province, grant numbers: 2021CXZX-357 and The Gansu Agricultural University College Students Scientific Research Training Program, grant numbers: 20180335.

Reviewers' comments:

Reviewer's Responses to Questions

**Comments to the Author**

1. Is the manuscript technically sound, and do the data support the conclusions?

Reviewer #1: Partly

Reviewer #2: Partly

2. Has the statistical analysis been performed appropriately and rigorously? 

Reviewer #1: Yes

Reviewer #2: I Don't Know

3. Have the authors made all data underlying the findings in their manuscript fully available?

Reviewer #1: Yes

Reviewer #2: Yes

4. Is the manuscript presented in an intelligible fashion and written in standard English?

Reviewer #1: Yes

Reviewer #2: Yes

5. Review Comments to the Author

Reviewer #1: PONE-D-23-06874

Expression and Role of Melatonin Membrane Receptors in the Hypothalamic-Pituitary-Testicular Axis of Tibetan Sheep in a Plateau Pastoral Area

The author in the present study worked on analyzing the expression of melatonin receptors (MTNR1A and MTNR1B) in different tissues of the HPTA axis. They also presented the characteristic difference between seasonal Tibetan sheep and non-seasonal Small Tail Han sheep. The conceptualization of the study and its execution is good, but I understand the manuscript requires an overhauling major revision in terms of design of certain experiments, conclusion obtained from the present data and presentation of results.

1. The authors have examined the morphological characteristics of tissues using H and E staining and electron microscopy. The biggest concern I have about this section is the detailed elaboration of morphological cell types and features presented are insufficiently supported due to limitation of these techniques for localization specially biomarker based.

Line 237-239 The epithelium of the epididymal duct consisted of principal cells, basal cells, halo cells, 238 pseudostratified columnar epithelium, and ciliated structures, and a large number of mature 239 spermatozoa could be observed in the epididymal duct.

It would be difficult and rather beyond the capacity of H\\E staining to differentiate the type of cells present in the epitheliums of the epididymal duct of testis tissue.

Line 243: Scanning electron microscopy showed that microglia were pancake-shaped and uniformly distributed in Tibetan sheep hypothalamic tissue

It’s difficult to conclude the shape of the highlighted cells marked as microglia and moreover in absence of immunolocalization its difficult to assess the type of cell.

Line 289: A. In the hypothalamus, MTNR1A and MTNR1B were mainly distributed around microglia (Fig. 4A1 and A2)…

Its difficult to assess the type of cells and morphological characteristics by the attached figures.

2. The author while analyzing the level of expression by immunofluorescence (Line 302) cannot compare the fluorescence intensities to evaluate expression among two different antibodies ( MTNR1A and MTNR1B). I suggest authors to remove this comparative account as many factors determine the fluorescence intensity: the affinity, avidity, clonality, types of epitopes chosen ….and stringency of washing ( if not taken into account) .

3. Mere presence of MT receptor in microglia cannot be the basis of concluding that microglia may have role in seasonal regulation of melatonin and specially when we have not even shown them by co-localization.

4. Concluding that MTNR1A and MTNR1B are the primary membrane receptors mediating the regulatory role of MT in the HPTA of sheep based on the present experiment is erroneous as no experiments were conducted to see the expression of other subtypes MTNR1C…

5. The author had present detailed comparative localization of difference in mRNA and proteins in HPTA tissues which is a good finding and may avoid morphological comparisons without solid experimental data.

Reviewer #2: The paper "Expression and Role of Melatonin Membrane Receptors in the Hypothalamic-Pituitary" by Yang et al. evaluated expression of MTNR1A and MTNR1B in the hypothalamic-pituitary-testicular axis (HPTA) of Tibetan sheep at different reproductive stages. The authors suggested that this study should help advance research on the mechanism of reproductive regulation of the HPTA in male animals and provide reference data for improving the reproductive rate of seasonal breeding animals. Although the question addressed in this paper could have been of interest for the scientific community, this manuscript needs of modifications.

The authors should provide an adequate introduction for their manuscript. Specifically, revise the second and fourth paragraphs. They are confusing. The hypothesis and aims of the study could be clearer.

For this reviewer, the Figure 1 does not improve the understanding of the text. It is suggested to delete.

Were slaughterhouse samples used? This can raise raises questions about the nutritional status of the animals and brings more individual variability. We recommend, if possible, to provide this information. In addition, it would be appropriate to indicate the season at which the samples were collected.

The material and methods section lacks several important methodological details; add reference for methodologies used. Although many journals seem to accept IHC results without what I and many others consider appropriate controls, this can be a concern. The absence of staining when the primary antibody is omitted is a control for nonspecific binding of the secondary antibody; this result is not evidence for the specificity of staining with the primary antibody. The proper negative control should be substitution of serum or isotype-specific immunoglobulins at the same protein concentration as the primary antibody. Was any positive control used?

Statistical analysis: Did the authors check the data for normal distribution?

I think part of the discussion needs improvement. I consider the lack of some lines that give greater importance to the work carried out, emphasizing its biological relevance and possible practical applications of the findings. Please expand on this.

6. PLOS authors have the option to publish the peer review history of their article (what does this mean?). If published, this will include your full peer review and any attached files.

Reviewer #1: No

Reviewer #2: No

---

## [Author Response · Author response to Decision Letter 0]

22 Jul 2023

Response to Reviewer 1

Dear editors and reviewers,

Thank you for kindly reviewing our manuscript (article number: PONE-D-23-06874) entitled “Expression and Role of Melatonin Membrane Receptors in the Hypothalamic-Pituitary-Testicular Axis of Tibetan Sheep in a Plateau Pastoral Area”, which we submitted to PLOS ONE for publication. We greatly appreciate your detailed and valuable comments and suggestions.

The revised manuscript has been carefully modified in accordance with the comments from the reviewers. These revisions have been marked using “tracked changes” in the revised manuscript. Point-by-point responses to the comments and suggestions are also provided below. The line numbers in the reply are confirmed by the revised manuscript.

Point 1:The authors have examined the morphological characteristics of tissues using H and E staining and electron microscopy. The biggest concern I have about this section is the detailed elaboration of morphological cell types and features presented are insufficiently supported due to limitation of these techniques for localization specially biomarker based. 

1.1 Line 237-239 The epithelium of the epididymal duct consisted of principal cells, basal cells, halo cells, 238 pseudostratified columnar epithelium, and ciliated structures, and a large number of mature 239 spermatozoa could be observed in the epididymal duct.

It would be difficult and rather beyond the capacity of H\\E staining to differentiate the type of cells present in the epitheliums of the epididymal duct of testis tissue. 

Response 1.1: Thank you for your valuable comment. We have taken your suggestion and removed this mischaracterization from the manuscript (L241-244).

1.2 Line 243: Scanning electron microscopy showed that microglia were pancake-shaped and uniformly distributed in Tibetan sheep hypothalamic tissue

It’s difficult to conclude the shape of the highlighted cells marked as microglia and moreover in absence of immunolocalization its difficult to assess the type of cell. 

Response 1.2: Thank you for your kind advice. Because the cells mainly distributed in the region of the supraoptic nucleus of the hypothalamus are microglia, and there are reports in the literature that microglia are ovoid or rounded, we deliberately selected tissues in the region of the supraoptic nucleus of the hypothalamus during scanning electron microscopy sampling, so the protruding ovoid cells in Fig. 3A were considered to be microglia by us. However, considering the objectivity of your valuable comments, we still revised the description of the manuscript to ensure the accuracy of the results of the article (L248-249).

1.3 Line 289: A. In the hypothalamus, MTNR1A and MTNR1B were mainly distributed around microglia (Fig. 4A1 and A2)… 

Its difficult to assess the type of cells and morphological characteristics by the attached figures.

Response 1.3: Thank you for your kind advice. We have revised the manuscript as required, and highlighted it in red font (L292-293).

Point 2:The author while analyzing the level of expression by immunofluorescence (Line 302) cannot compare the fluorescence intensities to evaluate expression among two different antibodies ( MTNR1A and MTNR1B). I suggest authors to remove this comparative account as many factors determine the fluorescence intensity: the affinity, avidity, clonality, types of epitopes chosen ….and stringency of washing ( if not taken into account) . 

Response 2: Thank you very much for your comments. We have redone the layout of Figure 5 and deleted the fluorescence intensity comparison according to your suggestion.

Point 3:Mere presence of MT receptor in microglia cannot be the basis of concluding that microglia may have role in seasonal regulation of melatonin and specially when we have not even shown them by co-localization. 

Response 3:Thank you for your kind suggestion. We have revised this conclusion in accordance with your comments, and have revised the discussion related to this conclusion against it in the full manuscript, and are highlighting it in red font (L420-424 and L445-449). 

Point 4:Concluding that MTNR1A and MTNR1B are the primary membrane receptors mediating the regulatory role of MT in the HPTA of sheep based on the present experiment is erroneous as no experiments were conducted to see the expression of other subtypes MTNR1C… 

Response 4:Thank you for the reminder. We have revised this part of the conclusion and highlighted it in red in the manuscript (L528-529).

Point 5:The author had present detailed comparative localization of difference in mRNA and proteins in HPTA tissues which is a good finding and may avoid morphological comparisons without solid experimental data.

Response 5:Thank you very much for your comments, and for recognizing our work!

Response to Reviewers 2

Dear editors and reviewers,

Thank you for kindly reviewing our manuscript entitled “Expression and Role of Melatonin Membrane Receptors in the Hypothalamic-Pituitary-Testicular Axis of Tibetan Sheep in a Plateau Pastoral Area”, which we submitted to Plos one for publication. We greatly appreciate your detailed and valuable comments and suggestions. Here are our responses to the questions you raised.

Reviewer #2: The paper "Expression and Role of Melatonin Membrane Receptors in the Hypothalamic-Pituitary" by Yang et al. evaluated expression of MTNR1A and MTNR1B in the hypothalamic-pituitary-testicular axis (HPTA) of Tibetan sheep at different reproductive stages. The authors suggested that this study should help advance research on the mechanism of reproductive regulation of the HPTA in male animals and provide reference data for improving the reproductive rate of seasonal breeding animals. Although the question addressed in this paper could have been of interest for the scientific community, this manuscript needs of modifications.

Point 1:The authors should provide an adequate introduction for their manuscript. Specifically, revise the second and fourth paragraphs. They are confusing. The hypothesis and aims of the study could be clearer.

Response 1: Thank you for your valuable comment. Following your suggestions we have revised the second and fourth paragraphs of the manuscript and added experimental hypotheses to the manuscript (L61-63 and L91-98).

Point 2:For this reviewer, the Figure 1 does not improve the understanding of the text. It is suggested to delete.

Response 2:Thank you for your kind suggestion. We have removed Figure1 from the manuscript (L70-71).

Point 3:Were slaughterhouse samples used? This can raise raises questions about the nutritional status of the animals and brings more individual variability. We recommend, if possible, to provide this information. In addition, it would be appropriate to indicate the season at which the samples were collected.

Response 3: Thank you for your question. Our samples were collected from a breeding cooperative in Datong District, Xining City, Qinghai Province. After a health status check, we selected healthy, well-nourished, disease-free adult rams and transported them to a designated slaughterhouse, where our team of veterinarians executed the rams by carotid artery bloodletting and subsequently collected samples. We have added relevant notes in the "Materials and Methods" section of the manuscript, as well as the time of sample collection (L108-110 and L112-114).

Point 4:The material and methods section lacks several important methodological details; add reference for methodologies used. Although many journals seem to accept IHC results without what I and many others consider appropriate controls, this can be a concern. The absence of staining when the primary antibody is omitted is a control for nonspecific binding of the secondary antibody; this result is not evidence for the specificity of staining with the primary antibody. The proper negative control should be substitution of serum or isotype-specific immunoglobulins at the same protein concentration as the primary antibody. Was any positive control used?

Response 4:Thank you for your kind suggestion. We have added references to the experimental methods in the manuscript. The negative control method you described is very correct, and our negative control experiments did use the same concentration of goat serum to replace the primary antibody, we just omitted these in the description. We have made changes in the experimental methods of the manuscript (L140-141).

Point 5:Statistical analysis: Did the authors check the data for normal distribution?

Response 5: Thank you for your valuable comment. Yes, we have tested the data for normal distribution before performing the "t" test, the software we used was SPSS 17.0, and the method used was the graphical test, and we found that all the data conformed to the characteristics of the normal distribution by plotting the histograms. We have made changes in the manuscript (L229-230).

Point 6:I think part of the discussion needs improvement. I consider the lack of some lines that give greater importance to the work carried out, emphasizing its biological relevance and possible practical applications of the findings. Please expand on this.

Response 6:Thank you for your kind advice. We have revised the manuscript as required (L420-424, L445-449, L493-496, and L522-526).

---

## [Decision Letter · Decision Letter 1]

16 Aug 2023

Expression and Role of Melatonin Membrane Receptors in the Hypothalamic-Pituitary-Testicular Axis of Tibetan Sheep in a Plateau Pastoral Area

PONE-D-23-06874R1

Dear Dr. Yang,

We’re pleased to inform you that your manuscript has been judged scientifically suitable for publication and will be formally accepted for publication once it meets all outstanding technical requirements.

Kind regards,

Martine Migaud

Academic Editor

PLOS ONE

Additional Editor Comments (optional):

The authors have satisfactorily addressed the concerns raised by the two reviewers. The manuscript is now sound and acceptable for publication.

Reviewers' comments:

Reviewer's Responses to Questions

**Comments to the Author**

1. If the authors have adequately addressed your comments raised in a previous round of review and you feel that this manuscript is now acceptable for publication, you may indicate that here to bypass the “Comments to the Author” section, enter your conflict of interest statement in the “Confidential to Editor” section, and submit your "Accept" recommendation.

Reviewer #1: All comments have been addressed

Reviewer #2: (No Response)

2. Is the manuscript technically sound, and do the data support the conclusions?

Reviewer #1: Yes

Reviewer #2: (No Response)

3. Has the statistical analysis been performed appropriately and rigorously? 

Reviewer #1: I Don't Know

Reviewer #2: (No Response)

4. Have the authors made all data underlying the findings in their manuscript fully available?

Reviewer #1: (No Response)

Reviewer #2: (No Response)

5. Is the manuscript presented in an intelligible fashion and written in standard English?

Reviewer #1: Yes

Reviewer #2: (No Response)

6. Review Comments to the Author

Reviewer #1: The authors have addressed most of the concern and manuscript presently is in much better shape and more congruent technically.

Reviewer #2: The authors addressed satisfactorily to the various points raised by this reviewer. This reviewer has no further comment.

7. PLOS authors have the option to publish the peer review history of their article (what does this mean?). If published, this will include your full peer review and any attached files.

Reviewer #1: No

Reviewer #2: No

---

## [Editor Report · Acceptance letter]

16 Oct 2023

PONE-D-23-06874R1 

Expression and Role of Melatonin Membrane Receptors in the Hypothalamic-Pituitary-Testicular Axis of Tibetan Sheep in a Plateau Pastoral Area 

Dear Dr. Yang:

I'm pleased to inform you that your manuscript has been deemed suitable for publication in PLOS ONE. Congratulations! Your manuscript is now with our production department. 

Kind regards, 

on behalf of

Dr. Martine Migaud 

Academic Editor

PLOS ONE